# Polymicrobial bloodstream infections a risk factor for mortality in neonates at the national hospital, Tanzania: A case-control study

**Joel Manyahi**[1]*, **Agricola Joachim**[1], **Frank Msafiri**[1], **Mary Migiro**[1], **Anthon Mwingwa**[1], **Mabula Kasubi**[2], **Helga Naburi**[3], **Mtebe Venance Majigo**[1]

**1** Department of Microbiology and Immunology, Muhimbili University of Health and Allied Sciences, Dar es Salaam, Tanzania, **2** Muhimbili National Hospital, Dar es Salaam, Tanzania, **3** Department of Pediatrics and Child Health, Muhimbili University of Health and Allied Sciences, Dar es Salaam, Tanzania

* manyahijoel@yahoo.com

**Data Availability Statement:** Minimal data have been included in the submission as a Supporting information.

## Abstract

### Background

Polymicrobial bloodstream infections (BSI) are difficult to treat since empiric antibiotics treatment are frequently less effective against multiple pathogens. The study aimed to compare outcomes in patients with polymicrobial and monomicrobial BSIs.

### Methods

The study was a retrospective case-control design conducted at Muhimbili National Hospital for data processed between July 2021 and June 2022. Cases were patients with polymicrobial BSI, and controls had monomicrobial BSI. Each case was matched to three controls by age, admitting ward, and duration of admission. Logistic regression was performed to determine independent risk factors for in-hospital and 30-day mortality.

### Results

Fifty patients with polymicrobial BSI and 150 with monomicrobial BSI were compared: the two arms had no significant differences in sex and comorbidities. The most frequent bacteria in polymicrobial BSI were *Klebsiella pneumoniae* 17% (17/100) and *Enterobacter* species 15% (15/100). In monomicrobial BSI, *S. aureus* 17.33% (26/150), *Klebsiella pneumoniae* 16.67% (25/150), and *Acinetobacter* species 15% (15/150) were more prevalent. Overall, isolates were frequently resistant to multiple antibiotics tested, and 52% (130/250) were multidrug resistance. The 30-day and in-hospital mortality were 33.5% (67/200) and 36% (72/200), respectively. On multivariable analysis, polymicrobial BSIs were independent risk factors for both in-hospital mortality (aOR 2.37, 95%CI 1.20–4.69, p = 0.01) and 30-day mortality (aOR 2.05, 95%CI 1.03–4.08), p = 0.04). In sub-analyses involving only neonates, polymicrobial BSI was an independent risk factor for both 30-day mortality (aOR 3.13, 95% CI 1.07–9.10, p = 0.04) and in-hospital mortality (aOR 5.08, 95%CI 1.60–16.14, p = 0.006). Overall, the median length of hospital stay post-BSIs was numerically longer in patients with polymicrobial BSIs.

**Funding:** The author(s) received no specific funding for this work.

**Competing interests:** The authors have declared that no competing interests exist.

## Conclusion

Overall, polymicrobial BSI was a significant risk for mortality. Patients with polymicrobial BSI stay longer at the hospital than those with monomicrobial BSI. These findings call for clinicians to be more aggressive in managing polymicrobial BSI.

## Background

Bloodstream infections (BSI) in low-middle-income countries (LMICs) are steadily associated with poor treatment outcomes, with the rising concern associated with antibiotic-resistant bacteria. BSI has been reported in 10–13% of inpatients with severe febrile illness in East and West Africa, with an estimated case fatality rate of 12% [1, 2].

Following the reduction of malaria, a frequent cause of febrile illness in Tanzania, BSIs remain a common cause of hospitalization in children with febrile illness. In Tanzania, BSIs is detected in 11–14% of children with severe febrile illness [3, 4]. Studies have shown that BSIs in adults and children result in worse outcomes, such as high mortality and length of hospital stay [5–8]. In addition, increasing evidence of antibiotic-resistant bacteria, including methicillin-resistant *Staphylococcus aureus* (MRSA), extended-spectrum beta-lactamases (ESBL), and multidrug-resistant (MDR) bacteria in BSIs, has been reported in Tanzania [3–5, 9].

Studies in Tanzania have reported that BSIs due to antibiotic-resistant bacteria had a significantly higher risk of death than those with antibiotic susceptible bacteria [5, 8, 9]. Mortality in children with ESBL BSI was approximately 2-fold higher than in non-ESBL septicemia [9]. Our recent study found that patients with MDR BSIs (aOR = 15) had a higher risk of dying than those with non-MDR BSIs [5]. However, studies still need to evaluate the outcomes of polymicrobial BSIs in Tanzania thoroughly.

Polymicrobial BSIs are frequently difficult to treat, and choosing the appropriate empiric antibiotics effective for more than one pathogen is more challenging. It is, therefore, vital to promptly identify the pathogens causing polymicrobial BSIs and administer the proper empiric antibiotics/drugs. In addition, studies on adults and children from other settings have demonstrated that mortality attributable to polymicrobial BSIs is higher than in monomicrobial BSIs [10–13]. Therefore, we carried out this retrospective case-control study at Muhimbili National Hospital (MNH) to ascertain the impact of polymicrobial BSIs on patient outcomes. A better understanding of the risk factors for worse outcomes in patients with BSIs would improve clinical outcomes for patients.

## Materials and methods

### Study setting and design

This was a retrospective case-control study for the data processed between July 2021 and June 2022 at MNH, Dar es Salaam, Tanzania. The data for this study were accessed and collected between July 1, 2022, and September 30, 2022. MNH is a tertiary-level hospital in Tanzania and serves as a teaching hospital for Muhimbili University of Health and Allied Sciences (MUHAS). It has a 1,500-bed capacity and attends up to 2,000 outpatients daily. The hospital has an infection control policy and an infection prevention and control (IPC) unit. Furthermore, blood culture sample in all wards is collected under aseptic condition and transferred to the laboratory within one hour of collection. Under ideal situations, the laboratory reporting system allows to flag clinicians with preliminary results immediately when the blood culture

flag-positive and final results 2–3 days after the blood culture is positive. In this retrospective case-control study, cases were patients with polymicrobial BSIs, and controls were patients with monomicrobial BSIs. Each case was matched to three controls by age ±5 years (neonates ±5 days), admission date, and admitted wards.

Blood culture results of patients aged 0–86 years with polymicrobial and monomicrobial growth were first retrieved from the laboratory registry. Patients' information, including ages, sex, admission wards, date of admission, time of diagnosis of BSIs, date of discharge, comorbidity, and clinical outcomes (death or survival), were collected from clinical case notes. All cases and their matched controls with complete demographics and clinical information were included in the analysis. Only the first episode was included in cases where patients had two or more blood cultures positive with similar organisms. Patients with incomplete demographics and medical records were excluded from the analysis.

## Microbiological methods for blood culture at MNH

Blood samples for culture were incubated in the BD BACTEC FX 200 at MNH, central pathology laboratory. Only one set of blood cultures was performed in adults and pediatrics. A set of BD BACTEC Plus Aerobic and Anaerobic/F Culture Vials (Becton Dickinson, BD) were processed for adults. For pediatrics, 1–3 mL of blood sample was collected in the BD BACTEC Peds PlusTM/F Culture Vials. Blood culture vials in BD BACTEC 200 were incubated for a maximum of five days before being regarded as a negative culture.

Positive blood cultures were Gram-stained and sub-cultured into sheep blood agar, chocolate agar, and MacConkey agar. Identification of bacteria was based on colonial morphology, Gram stain, and biochemical tests, including conventional API20E (BioMérieux, Marcy, I'Etoile, France) and Staphaurex (Remel Europe Ltd, Dartford, UK). Antimicrobial susceptibility testing was performed and interpreted according to clinical and laboratory standard institute (CLSI) guidelines [14].

## Definition of terms

Polymicrobial BSI was defined as bacteremia with at least two different organisms from the same blood culture. The onset of BSI was defined as the day when the blood was drawn for culture that subsequently produced positive pathogenic growth. Laboratory-confirmed BSIs was defined as blood culture with the growth of recognizable pathogens in patients with signs and symptoms of BSIs. Coagulase-negative staphylococci were considered a pathogen in immunosuppressed patients and neonates, in accordance with hospital blood culture standard operational procedure. *Micrococcus*, *Corynebacterium*, *Viridans* Group *streptococci*, and *Bacillus* species were considered as contaminants. The 30-day mortality was defined as death from any cause within 30 days of the onset of BSI. In-hospital mortality was defined as death from any cause during index hospitalization following diagnosis of BSI. The length of hospital stay was calculated from the date of diagnosis of BSIs. It excluded all patients with in-hospital mortality.

In addition, appropriate treatment was defined as empirical antimicrobial therapy with an antibiotic that has in vitro activity against the isolated pathogen based on susceptibility testing results.

## Statistical analysis

Categorical variables were presented as frequency and percentages. A comparison of categorical variables between cases and controls was performed using Pearson chi-square. The median length of hospital stays among cases, and controls was compared using the Wilcoxon rank-

sum test. Binary logistic regression compared outcomes of interest (30-day and in-hospital mortality) between cases and controls. All plausible biological variables (Admission ward, comorbidity, and appropriate treatment) with *p*-values < 0.20 on the bivariate analysis were included in the logistic regression modal for the multivariable analysis.

Multivariable logistic regression was performed to find an independent association of polymicrobial and monomicrobial BSIs to 30-day and in-hospital mortality. We adjusted for age, sex, admission ward, appropriate treatment, and comorbidity as potential confounders, defining confounders as factors that change the effect size by $\geq$10%.

We performed a sub-analysis for neonates and non neonates, where bivariate and multivariable logistic regression were performed. Independent associations between polymicrobial BSIs and outcomes of interest (30-day and in-hospital mortality) were adjusted for age, sex, admission ward, and comorbidity. All statistical analysis was performed using STATA version 16 (College Station, TX). A significance level of $\leq$0.05 was used, and all p-values refer to two-sided tests.

## Ethics approval and consent to participate

Ethical approval was granted by the Muhimbili University of Health and Allied Sciences senate and publications committee Ref. No DA.25/111/01C/157. Permission to conduct the study was approved by Muhimbili National Hospital management. As the study was a retrospective study using identified patients, informed consent was waived by Muhimbili University of Health and Allied Sciences senate and publication committee. All methods were carried out under relevant guidelines and regulations in the declaration.

## Results

### Demographics and clinical characteristics

Between July 2021 and June 2022, 3398 blood cultures were processed at the microbiology laboratory of MNH. Overall, the proportion of BSI was 19% (642/3398). From 642 patients with BSI, we obtained 50 cases with polymicrobial BSI, matched to 150 controls with monomicrobial BSI in a ratio of 1:3. The age distributions among the cases and controls ranged from 1 day to 86 years, with majority of partipants being neonates. There were no significant differences in sex and comorbidities between polymicrobial BSI and monomicrobial BSI (Table 1).

**Table 1. Demographic-clinical characteristics of the participants.**

| Variable | Monomicrobial BSIs(N = 150) | Polymicrobial BSIs (N = 50) | *p*-value |
|---|---|---|---|
| Age (median, IQR) | 1 (1–40) | 1 (1–40) | |
| <1month | 69 | 23 | |
| <1 year | 18 | 6 | |
| 2–16 year | 9 | 3 | |
| >17 years | 54 | 18 | |
| Sex, Male | 75 (50%) | 26 (52%) | 0.8 |
| Ward, ICU | 90 (60%) | 31 (62%) | 0.8 |
| Comorbidity, Yes | 68 (45.3%) | 22 (44%) | 0.87 |
| Appropriate treatment, Yes | 72 (48%) | 18 (36%) | 0.14 |

IQR- Interquartile ratio; ICU-intensive care unit

**Table 2. Bacterial isolates from monomicrobial and polymicrobial bloodstream infections.**

| Bacteria | Monomicrobial BSI N(%) | Polymicrobial BSI N(%) |
|---|---|---|
| *K. pneumoniae* | 25 (16.67) | 17 (17.00) |
| *E. coli* | 17(11.33) | 7 (7.00) |
| *Enterobacter* spp | 13(8.67) | 15(15.00) |
| *Other Enterobacterales | 13 (8.67) | 16 (16.00) |
| *Acinetobacter* spp | 15 (15) | 9 (9.00) |
| *P. aeruginosa* | 14 (9.33) | 8 (8.00) |
| #Non-Enterobacterales | 4 (2.67) | 6(6.00) |
| *S. aureus* | 26 (17.33) | 9(9.00) |
| CoNS | 20 (13.33) | 9(9.00) |
| *Streptococcus* spp | 2(1.33) | 3(3.00) |
| *Candida* spp | 1 (0.67) | 1 (1.00) |
| **Total** | **150** | **100** |

*Other Enterobacterales: *Morganella morganii*, *Serratia mercescens*, *Providencia* spp, *Citrobacter* species

#Non-Enterobacterales: *Aeromonas* spp, *Chromobacteria violaceum*; CoNS:Coagulase negative Staphylococcus

## Microbiology of bloodstream infections

Table 2 provides the microbiology for both polymicrobial and monomicrobial BSIs. The bacteria-causing polymicrobial and monomicrobial BSIs were similar, with the predominance of Gram-negatives, although they varied in frequency. The most frequent bacteria in polymicrobial BSIs were *Klebsiella pneumoniae* 17% (17/100) and *Enterobacter* species 15% (15/100). Combinations of these two bacteria, as well as *S. aureus*/*Pseudomonas aeruginosa* and *Acinetobacter* species/CoNS, were also frequent, occurring in three instances each. In monomicrobial BSIs, *S. aureus* 17.33% (26/150), *Klebsiella pneumoniae* 16.67% (25/150), and *Acinetobacter* species 10% (15/150) were more prevalent (Table 2).

## Antimicrobial susceptibility pattern

Overall, isolates were frequently resistant to multiple antibiotics tested, and 53% (130/243) were multidrug resistance. We observed that 77% (27/35) of *S. aureus* and 85% (25/29) of coagulase-negative Staphylococcus were resistant to cefoxitin, hence methicillin-resistant strains. All Enterobacterales were frequently resistant to all antibiotics tested except for meropenem and amikacin. Furthermore, *S. aureus* was less resistant to chloramphenicol (28%), clindamycin (38%), and trimethoprim-sulfamethoxazole (45%) (Table 3).

## Outcomes in bloodstream infections

Overall, the 30-day mortality was 33.5% (67/200). The risk of mortality was higher in polymicrobial BSI compared to monomicrobial BSI, with an odd ratio (OR) of 1.83 (95% confidence interval (CI) 0.95–3.54), though the difference was not statistically significant (p = 0.07). However, after adjusting for possible confounders including sex, age, the admission ward, appropriate treatment, and comorbidity in a multivariable model, polymicrobial BSI independently predicted 30-day mortality, aOR 2.05, 95%CI 1.03–4.08), p = 0.04. Additionally, in-hospital mortality was 36% (72/200), with cases dying more frequently than controls (50% vs. 31.3%). We observed that polymicrobial BSI was a risk factor for in-hospital death based on univariable analysis crude odd ratio (cOR), 2.19, 95%CI 1.14–4.21, p = 0.02. We considered sex, age, the admission ward, appropriate treatment, and comorbidity in the final multivariable analysis

**Table 3. Antimicrobial susceptibility patterns of bacteria from bloodstream infections.**

| Bacteria (n) | AMP | AMC | TZP | CRO | CAZ | FEP | MEM | AMK | CN | CIP | CHL | SXT | P | FOX | ERY | CLI |
|---|---|---|---|---|---|---|---|---|---|---|---|---|---|---|---|---|
| | | | | | | | Percent Resistance | | | | | | | | | |
| *K. pneumoniae* (42) | 100 | 80 | 75 | 72 | 71 | 76 | 44 | 39 | 61 | 69 | 68 | 59 | | | | |
| *E. coli* (24) | 92 | 81 | 75 | 80 | 74 | 90 | 26 | 44 | 53 | 76 | 36 | 100 | | | | |
| *Enterobacter* spp (28) | 100 | 89 | 60 | 76 | 81 | 74 | 79 | 36 | 70 | 79 | 69 | 70 | | | | |
| *Other Enterobacterales (29) | 84 | 75 | 38 | 84 | 52 | 87 | 45 | 38 | 60 | 58 | 53 | 69 | | | | |
| *Acinetobacter* spp (24) | | | 70 | 89 | 59 | 68 | 57 | 37 | 65 | 50 | 46 | 55 | | | | |
| *P. aeruginosa* (22) | | | 58 | | 39 | 31 | 24 | 32 | 58 | 47 | | | | | | |
| #Non-Enterobacterales (10) | | | 63 | | 33 | 44 | 0 | 22 | 56 | 22 | 14 | 67 | | | | |
| *S. aureus* (35) | | | | | | | | | 60 | 58 | 28 | 45 | 100 | 77 | 80 | 38 |
| CoNS (29) | | | | | | | | | 62 | 94 | 16 | 90 | 100 | 85 | 89 | 90 |
| **Overall** | **91** | **79** | **64** | **80** | **64** | **72** | **43** | **36** | **62** | **64** | **48** | **67** | **100** | **81** | **84** | **64** |

AMK-Amikacin, AMC-Amoxycillin-clavulanic acid, AMP-Ampicillin, FOX-Cefoxitin, CAZ-Ceftazidime, CRO-Ceftriaxone, CHL-Chloramphenicol, CIP-Ciprofloxacin, CLI-Clindamycin, ERY-Erythromycin, GEN-Gentamicin, MEM-Meropenem, TZP-Tazobactam-piperacillin, P-Penicillin, SXTTrimethoprim-sulfamethoxazole. Other Enterobacterales: *Morganella morganii*, *Serratia mercescens*, *Providencia* spp, *Citrobacter* species
#Non-Enterobacterales: *Aeromonas* spp, *Chromobacteria violaceum*. CoNS:Coagulase negative Staphylococcus

model as potential confounding variables. After adjusting for these variables, we found that polymicrobial BSIs adjusted odd ratio (aOR) 2.37, 95% CI 1.20–4.69, p = 0.01, had an independent association with in-hospital mortality (Table 4).

In sub-analyses involving only neonates, we found that polymicrobial BSI had an independent association with both 30-day mortality (aOR 4.02, 95% CI 1.32–12.32, p = 0.01) and in-hospital mortality (aOR 6.18, 95% CI 1.89–20.22, p = 0.003) after adjusting for, sex, appropriate treatment, and comorbidities. For all age groups and neonates, the median length of hospital stay post-BSIs was numerically longer in patients with polymicrobial than in those with monomicrobial BSIs, though the difference was not statistically significant (Table 5).

In a sub-analysis of non-neonates, we observed that polymicrobial BSI had no association with both in-hospital and 30-day mortality (Table 5).

## Discussion

Our study demonstrates that polymicrobial BSIs are significantly associated with in-hospital mortality compared to monomicrobial BSIs in patients of all age groups. We observed

**Table 4. Multivariable analysis assessing the association between BSIs and mortality.**

| Outcome | Monomicrobial BSIs | Polymicrobial BSIs | cOR, 95%CI, *p*-value | aOR, 95%CI, *p*-value |
|---|---|---|---|---|
| **All patients** | **N = 150** | **N = 50** | | |
| 30-days mortality (%) | 30 | 44 | 1.83 (0.95–3.54), 0.07 | 2.05, (1.03–4.08), 0.04 |
| In-hospital mortality (%) | 31.3 | 50 | 2.19 (1.14–4.21), 0.02 | 2.37, (1.20–4.69), 0.01 |
| **Neonates only** | **N = 69** | **N = 23** | | |
| 30-days mortality (%) | 42.03 | 65.22 | 2.78, 1.04–7.39, 0.04 | 4.02, 1.32–12.23, 0.01 |
| In-hospital mortality (%) | 42.03 | 73.91 | 4.20, 1.48–11.92, 0.007 | 6.18, 1.89–20.22, 0.003 |
| **Non-neonates** | **N = 81** | **N = 27** | | |
| 30-days mortality (%) | 19.75 | 25.93 | 1.42, 0.51–3.94, 0.50 | 1.55, 0.55–4.41, 0.59 |
| In-hospital mortality (%) | 22.22 | 29.63 | 1.47, 0.55–3.91, 0.45 | 1.59, 0.58–4.36, 0.36 |

BSI: Bloodstream infections, cOR: crude odd ratio, aOR: adjustable odd ratio, CI: Confidence interval

**Table 5. Length of hospital stay post-bloodstream infections.**

| Outcome | Monomicrobial BSIs | Polymicrobial BSIs | p-value |
|---|---|---|---|
| **All patients** | **N = 105** | **N = 28** | |
| LOS (median, IQR) | 13 (5–24) | 26 (6–37) | 0.2 |
| **Neonates only** | **N = 43** | **N = 8** | |
| LOS (median, IQR) | 13 (6–32) | 32 (21.5–62) | 0.07 |

LOS: Length of hospital stay, IQR: Interquartile range, BSI: Bloodstream infections

significantly higher 30-day mortality in participants with polymicrobial BSI than those with monomicrobial BSI. We also found that patients with polymicrobial BSIs had a longer median length of hospital stay than patients with monomicrobial BSIs, but the differences were not statistically significant. A similar finding has been reported from previous studies, which found polymicrobial BSIs increase the risk of mortality [11, 15]. The difficulties in selecting appropriate empiric antimicrobials targeting multiple pathogens and delays in reporting blood culture results could impact our study's observed findings.

Most participants in this study received inappropriate treatment, which was more common in the polymicrobial BSI arm; however, we adjusted for inappropriate treatment in our analysis, and polymicrobial remained the main contributor to mortality. We could not account for the turn-around time of reporting polymicrobial BSI bacteriology results to clinicians as a possible contributor to the delay in initiation of treatment, possibly leading to increased mortality. Subsequently, ceftriaxone was the most commonly prescribed empiric antibiotic on review of clinical case notes, and most bacteria we isolated were resistant to this antibiotic. Our findings call for microbiologists and clinicians to be more vigilant when blood culture reveals polymicrobial BSI. We noted that the blood culture standard operating procedure does not detail how to communicate when polymicrobial BSIs are detected. On the other hand, the guidelines for treating sepsis consider only aerobic and anaerobic polymicrobial BSIs, and consideration is not given to aerobic-aerobic polymicrobial BSIs.

When we examined the neonates' risk factors for 30-day mortality, we found polymicrobial BSIs were an independent risk for mortality in neonates after adjusting for all possible confounders. The previous study in other settings has observed similar findings in neonates, where polymicrobial bloodstream infections were associated with 3 fold increased risk of mortality compared to monomicrobial bloodstream infection [10]. However, other studies in neonates have found comparable 30-day mortality between polymicrobial and monomicrobial BSIs [16, 17]. Immature immunity in neonates increases vulnerability to serious infections leading to an increased mortality risk. On the other hand, inappropriate antibiotic therapy compounded by overwhelming virulence from more than one pathogen would have led to the severity of illness in neonates, leading to the worse clinical outcomes observed.

Studies have reported that polymicrobial infections are associated with forming polymicrobial biofilms, which tend to increase pathogens' virulence, host immune evasion, and antibiotic resistance, hence poor treatment outcomes [18–20]. Polymicrobial BSI occurs more in neonates with severe underlying conditions and prolonged indwelling central venous catheters [16]. As central venous catheters were not commonly used at MNH during study time, the observed cases can be attributed to overwhelming illness.

We found that the median length of hospital stay was longer in patients with polymicrobial BSIs than in patients with monomicrobial BSIs. This could be due to the ineffectiveness of empiric antibiotic therapy since it was impossible for the prescribed empiric antibiotics to be

effective in two different organisms. Therefore, the likelihood of receiving inappropriate treatment resulting in failure and prolonged hospital stays was obvious. Our findings are similar to those reported in the studies conducted in other settings [13, 21–23]. This finding calls for establishing a local cumulative antibiogram to guide empiric therapy in suspected bacteria infections, as it currently not existing in health facilities in the country.

The microbiology of polymicrobial BSIs was similar to that of monomicrobial infections in this study, with the predominant Gram-negative bacteria. This indicates that the sources for polymicrobial and monomicrobial BSIs are the same. Other studies examining the microbiological profiles of polymicrobial and monomicrobial BSIs have reported similar microbiology patterns in both cases [12, 17, 24]. However, depending on the variability of the population studied, the predominance of either Gram-positive or Gram-negative has been reported. On the other hand, we found that most bacteria causing BSIs in our study were MDR, which limits the available treatment options in our settings. The findings of this study emphasize the need for microbiology laboratories to promptly perform and report antimicrobial susceptibility results to guide evidence-based antibiotics therapy.

One of the important caveats of our study was that it was a retrospective design dependent on recorded information from clinical case notes. Information like the severity of BSIs and the classification of BSIs (community onset versus hospital onset) was not documented in the clinical case notes, and this means missing data could introduce information bias. However, missing end-point outcomes would be non-differentially distributed evenly across polymicrobial and monomicrobial BSIs. Furthermore, we performed multivariable analysis to control for possible confounding variables associated with mortality. Another important limitation of our study is that we could not assess the risk factors for polymicrobial BSIs because of the missing information. Understanding risk factors for polymicrobial BSIs would be critical to controlling and preventing infections before they even occur. However, our findings call for microbiologists and clinicians to manage polymicrobial BSIs because of the associated risk aggressively.

## Conclusion

In neonates, polymicrobial BSI is an independent risk for 30-day and in-hospital mortality. Overall, the median length of hospital stay was longer in patients with polymicrobial BSI than those with monomicrobial BSI. These findings highlight the need for clinicians to be more aggressive when polymicrobial BSI is detected. The laboratory should also prompt notify clinician when polymicrobial growth is detected in blood culture.

## Supporting information

**S1 Data.**
(XLSX)

## Acknowledgments

The authors acknowledge the members of the bacteriology section at the central pathology laboratory, MNH.

## Author Contributions

**Conceptualization:** Joel Manyahi.

**Data curation:** Mary Migiro, Anthon Mwingwa.

**Formal analysis:** Joel Manyahi.

**Investigation:** Joel Manyahi.

**Methodology:** Joel Manyahi.

**Writing – original draft:** Joel Manyahi.

**Writing – review & editing:** Agricola Joachim, Frank Msafiri, Mary Migiro, Mabula Kasubi, Helga Naburi, Mtebe Venance Majigo.

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
