## [Decision Letter · Decision Letter 0]

23 Oct 2023

PONE-D-23-28734Polymicrobial bloodstream infections a risk factor for mortality at the national hospital, Tanzania: A case-control studyPLOS ONE

Dear Dr. Manyahi,

Thank you for submitting your manuscript to PLOS ONE. After careful consideration, we feel that it has merit but does not fully meet PLOS ONE’s publication criteria as it currently stands. Therefore, we invite you to submit a revised version of the manuscript that addresses the points raised during the review process.

Please address all points raised by the reviewers and comments made directly on the manuscript. 

We look forward to receiving your revised manuscript.

Kind regards,

Iddya Karunasagar

Academic Editor

PLOS ONE

Additional Editor Comments:

Both reviewers have raised a number of points in data analysis and presentation that needs improvement. Please address all comments point by point.

Reviewers' comments:

Reviewer's Responses to Questions

**Comments to the Author**

1. Is the manuscript technically sound, and do the data support the conclusions?

Reviewer #1: Partly

Reviewer #2: Partly

2. Has the statistical analysis been performed appropriately and rigorously? 

Reviewer #1: Yes

Reviewer #2: No

3. Have the authors made all data underlying the findings in their manuscript fully available?

Reviewer #1: No

Reviewer #2: No

4. Is the manuscript presented in an intelligible fashion and written in standard English?

Reviewer #1: Yes

Reviewer #2: Yes

5. Review Comments to the Author

Reviewer #1: The evidence for poly-microbial BSI is not very clear just isolation of two or more bacteria from blood culture does not substantiate the the poly-microbial BSI.

Line 164 and 165 Acinetobacter and CONS, Pseudomonas and S.aureus in these combinations CONS and S.aureus could be skin contaminants in which case BSI is mono-microbial only. Therefore selection criteria of poly-microbial BSI cases to be clearly defined in which clinical conditions or comorbidities poly-microbial BSI is a possibility only such case to be selected.

Line 166 and 167 Acinetobacter 15% (15/150) calculation error.

While mentioning the antibiotic resistance consider the antibiotics used for blood stream infections not clindmycin, cotrimoxazole and chloramphenicol for S.aureus.

Which are the disease comorbidities were considered for statistical analysis because these disease comorbidities will play very significant role in hospital mortality and 30 day mortality in patients with BSI besides the antimicrobial resistance.

Reviewer #2: The paper made an interesting reading, dealing with polymicrobial infections Vs Monomicrbial Infections , the microbiological and clinical impact of these with respect to 30 day mortality and the extended hospital stay etc.

Comments for the same are attached , the authors are advised to kindly refer to the same and address all the points mentioned

6. PLOS authors have the option to publish the peer review history of their article (what does this mean?). If published, this will include your full peer review and any attached files.

Reviewer #1: No

Reviewer #2: No

---

## [Author Response · Author response to Decision Letter 0]

8 Dec 2023

Reviewer #1: The evidence for polymicrobial BSI is not very clear just isolation of two or more bacteria from blood culture does not substantiate the polymicrobial BSI.

Comment: Line 164 and 165 Acinetobacter and CONS, Pseudomonas and S.aureus in these combinations CONS and S.aureus could be skin contaminants in which case BSI is mono-microbial only. Therefore, selection criteria of polymicrobial BSI cases to be clearly defined in which clinical conditions or comorbidities polymicrobial BSI is a possibility only such case to be selected.

Response: Thank you for the comment. Our laboratory SoP for a blood culture states very clearly when to consider CoNS, and other possible skin contaminants as true pathogens, and we strictly adhered to the laboratory SoP in defining pathogens. S. aureus in blood culture is always considered a pathogen in our interpretation of the blood culture positive

Comment: Line 166 and 167 Acinetobacter 15% (15/150) calculation error.

Response: We appreciate for the comment, we have addressed calculation error

Comment: While mentioning the antibiotic resistance consider the antibiotics used for blood stream infections not clindamycin, cotrimoxazole and chloramphenicol for S. aureus.

Response: We appreciate for the comment. However, our results show the susceptibility pattern based on laboratory testing and not based treatment of S. aureus blood stream infections. In reporting AST results to clinician, our laboratory considers both bacteria and site of infections.

Comment: Which are the disease comorbidities were considered for statistical analysis because these disease comorbidities will play very significant role in hospital mortality and 30-day mortality in patients with BSI besides the antimicrobial resistance.

Response: Unfortunately, clinical conditions were so diverse, and a breakdown presentation was difficulties. Therefore, in our analysis we categorized either presence of comorbidity or not. 

Reviewer #2: The paper made an interesting reading, dealing with polymicrobial infections Vs Monomicrobial Infections , the microbiological and clinical impact of these with respect to 30 day mortality and the extended hospital stay etc.

Comments for the same are attached , the authors are advised to kindly refer to the same and address all the points mentioned

The paper deals with the differences (Both Microbiological yield and clinical outcomes) in patients with Polymicrobial and Monomicrobial Bacteremia Infections; The Following observations are in order and must be addressed before the manuscript can be taken forward 

Comment: The title of the paper does not reveal that the major component of the study is in children, adolescents, neonates, infants or adults. However, the Table No 4 & 5 refer to clinical outcomes in neonates only.

Response: Thank you for the comment. We agree that the title does not mention the major components of the participants. Our study aimed at creating awareness of the negative effects of polymicrobial bloodstreams on patient care for both laboratory personnel and clinicians. Being a lab-based retrospective, we therefore enrolled all patients with polymicrobial bloodstreams matching their counterparts by age, date of admission, and admitting wards. However, because we had a significant number of participants who were neonates, we decided to perform a sub-analysis for neonates only, as appears in Tables 4 and 5. On the other hand, we have done sub-analysis in none neonates, Therefore, we have modified the title to show the component affected.

Comment: Moreover, the Table – 1 which deals with age and sex distribution does not give a breakdown of the age groups and refers to a median age alone. 

Response: Thank you for the comment. Our study design was a case control, which was matched by age; therefore, we found most participants were populated in the same age category. However, we have added age breakdown in table1

Comment: The discussion section refers to certain observations which have been compared with other studies and these deal with outcomes in neonates only. 

Response: We appreciate for the comment, we have improved our citation and omitted references which do not match with our study population

Comment: In the section “Materials and Methods “ “study setting and design ‘ the authors have stated that results are obtained in age groups from 1-86 years and of these 50 patients were selected and 150 controls. What was the age distribution taken into consideration when this selection was made. The clinical outcomes may be different in different age groups with polymicrobial bacteremia. 

Response: We appreciate for the good comment, in this study we included all age groups from 0-86. Understanding the age have influence on the clinical outcome, first we adjusted for age in multivariable analysis. Then we did stratification analysis for neonate only and non-neonate. Having this analysis, we are assured we have controlled the age a possible confounder influencing our outcome of interest. 

Comment: The authors have stated a definition for Polymicrobial infections in the section “Definition of terms”. However, the term polymicrobial need not be restricted to 2 organisms as there are instances of blood stream infections caused by more than 2 organisms, including a yeast. How has the selection for polymicrobial infections data been made. Moreover the Results section talks of common organisms as Nos/ 100 which means that the authors have considered only 2 organisms per blood culture in a patient (total no of patients being 50). 

Response: Thank you for this observation. We completely agree polymicrobial infections can be caused by two or more bacteria. Unfortunate, during the review of the laboratory data, we did not come to an instance of having more than two pathogens. Furthermore, we had one case of polymicrobial bloodstream infections involving bacteria and candida species.

Comment: The clinical outcomes for blood stream infections in conditions such as Perforation peritonitis, Carcinoma colon etc may depend on the nos of organisms causing the Polymicrobial Infection. Has this been taken into account. There is no mention anywhere in the manuscript and yeasts do not form part of the study. 

Response: In our analysis we considered co-morbidities as one of the confounders, which could influence our outcome of interest, and in multivariable analysis we controlled for comorbidities, however, we found this did not influence our outcome. Furthermore, we did not document any case with perforation or peritonitis, but there was only one case of polymicrobial infections in patient with rectal carcinoma. In table 2, we have mentioned yeast being part of the study. 

Comment: The authors have stated that Viridans streptococcus and Corynebacteria have been disregarded as contaminants. However, these organisms may assume huge clinical significance in certain clinical situations such as immunocompromise and malignancies in patients with BSI’s. The authors have not provided a break up of clinical conditions in the 50 cases and 150controls which in itself may skew the clinical outcome data of patients. 

Response: Thank you for the comment. Our laboratory SoP for a blood culture states very clearly when to consider CoNS, Viridans Streptococcus, Corynebacterium as true pathogens, and we strictly followed the lab SoP regarding these as pathogenic. Unfortunately, clinical conditions were diverse, and a breakdown presentation in a table would be difficult for a reader.

Comment: The description of Results section under “Antimicrobial Susceptibility Pattern” does not match the data outlined in Table -3. 

Response: Thank you for observation, we have edited on the total number of MDR bacteria

Comment: There is no reference to certain Resistance mechanisms such as Carbapenem resistant Enterobacterales (CRE), Amp C enzymes MLSBi/c detection in S. aureus. All of these do impact treatment outcomes. How have these been factored in the selection of 50 cases and 150 controls in the study as these would most certainly affect the final analyses 

Response: Thank you for the valid comment. In the selection of cases and controls, we did not consider if patients had resistant strains or not because our study hypothesized that polymicrobial bloodstream infections were associated with poor treatment outcomes, and we designed this study as a case control to answer our research question. We had thought of controlling for MDR or resistant strains, but the analysis could have been different and not for a case-control study, maybe cross-sectional.

Comment: The section on Discussion gives vague references to 30-day mortality and extended hospital stay etc without actual figures and how these may compare with similar data brought out in other studies. There are equal references to other studies in neonates without actually discussing the distribution of cases among neonates in the present study. 

Response: Thank you for the comments, we have improved our reference to include reference from similar study group.

Comment: Based on the data presented and discussed the authors may have to modify the conclusion section as no definite conclusions emerge out of the discussion cited. 

Response: Thank you, we have rephrased our conclusion.

---

## [Decision Letter · Decision Letter 1]

28 Feb 2024

PONE-D-23-28734R1Polymicrobial bloodstream infections a risk factor for mortality in neonates at the national hospital, Tanzania: A case-control studyPLOS ONE

Dear Dr. Manyahi,

Thank you for submitting your manuscript to PLOS ONE. After careful consideration, we feel that it has merit but does not fully meet PLOS ONE’s publication criteria as it currently stands. Therefore, we invite you to submit a revised version of the manuscript that addresses the points raised during the review process.

Please address all comments of the reviewer and please indicate the changes in your response.==============================

We look forward to receiving your revised manuscript.

Kind regards,

Iddya Karunasagar

Academic Editor

PLOS ONE

Additional Editor Comments:

Manuscript still needs improvement as per reviewer comments.

Reviewers' comments:

Reviewer's Responses to Questions

**Comments to the Author**

1. If the authors have adequately addressed your comments raised in a previous round of review and you feel that this manuscript is now acceptable for publication, you may indicate that here to bypass the “Comments to the Author” section, enter your conflict of interest statement in the “Confidential to Editor” section, and submit your "Accept" recommendation.

Reviewer #3: (No Response)

2. Is the manuscript technically sound, and do the data support the conclusions?

Reviewer #3: Partly

3. Has the statistical analysis been performed appropriately and rigorously? 

Reviewer #3: Yes

4. Have the authors made all data underlying the findings in their manuscript fully available?

Reviewer #3: Yes

5. Is the manuscript presented in an intelligible fashion and written in standard English?

Reviewer #3: Yes

6. Review Comments to the Author

Reviewer #3: The authors need to mark ALL the sentences in the manuscript where they have responded to the reviewer's comments. This has not been done.

There are still some grammatical errors that need correction.

7. PLOS authors have the option to publish the peer review history of their article (what does this mean?). If published, this will include your full peer review and any attached files.

Reviewer #3: No

---

## [Author Response · Author response to Decision Letter 1]

5 Mar 2024

Reviewer #3: The authors need to mark ALL the sentences in the manuscript where they have responded to the reviewer's comments. This has not been done.

There are still some grammatical errors that need correction.

Response: Thank you for the comments to improve our manuscript. We have added yellow highlights in areas we have made changes in a Revised Manuscript with Track Changes. Furthermore, we have addressed some of the grammatical errors as suggested.

In addition, below are authors response from the previous reviewers

Response to review

Review of the paper titled Polymicrobial Blood stream Infections; a risk factor for mortality at the National Hospital Tanzania; A case control study 

Reviewer #1: The evidence for polymicrobial BSI is not very clear just isolation of two or more bacteria from blood culture does not substantiate the polymicrobial BSI.

Comment: Line 164 and 165 Acinetobacter and CONS, Pseudomonas and S.aureus in these combinations CONS and S.aureus could be skin contaminants in which case BSI is mono-microbial only. Therefore, selection criteria of polymicrobial BSI cases to be clearly defined in which clinical conditions or comorbidities polymicrobial BSI is a possibility only such case to be selected.

Response: Thank you for the comment. Our laboratory SoP for a blood culture states very clearly when to consider CoNS, and other possible skin contaminants as true pathogens, and we strictly adhered to the laboratory SoP in defining pathogens. S. aureus in blood culture is always considered a pathogen in our interpretation of the blood culture positive

Comment: Line 166 and 167 Acinetobacter 15% (15/150) calculation error.

Response: We appreciate for the comment, we have addressed calculation error

Comment: While mentioning the antibiotic resistance consider the antibiotics used for blood stream infections not clindamycin, cotrimoxazole and chloramphenicol for S. aureus.

Response: We appreciate for the comment. However, our results show the susceptibility pattern based on laboratory testing and not based treatment of S. aureus blood stream infections. In reporting AST results to clinician, our laboratory considers both bacteria and site of infections.

Comment: Which are the disease comorbidities were considered for statistical analysis because these disease comorbidities will play very significant role in hospital mortality and 30-day mortality in patients with BSI besides the antimicrobial resistance.

Response: Unfortunately, clinical conditions were so diverse, and a breakdown presentation was difficulties. Therefore, in our analysis we categorized either presence of comorbidity or not. 

Reviewer #2: The paper made an interesting reading, dealing with polymicrobial infections Vs Monomicrobial Infections , the microbiological and clinical impact of these with respect to 30 day mortality and the extended hospital stay etc.

Comments for the same are attached , the authors are advised to kindly refer to the same and address all the points mentioned

The paper deals with the differences (Both Microbiological yield and clinical outcomes) in patients with Polymicrobial and Monomicrobial Bacteremia Infections; The Following observations are in order and must be addressed before the manuscript can be taken forward 

Comment: The title of the paper does not reveal that the major component of the study is in children, adolescents, neonates, infants or adults. However, the Table No 4 & 5 refer to clinical outcomes in neonates only.

Response: Thank you for the comment. We agree that the title does not mention the major components of the participants. Our study aimed at creating awareness of the negative effects of polymicrobial bloodstreams on patient care for both laboratory personnel and clinicians. Being a lab-based retrospective, we therefore enrolled all patients with polymicrobial bloodstreams matching their counterparts by age, date of admission, and admitting wards. However, because we had a significant number of participants who were neonates, we decided to perform a sub-analysis for neonates only, as appears in Tables 4 and 5. On the other hand, we have done sub-analysis in none neonates, Therefore, we have modified the title to show the component affected.

Comment: Moreover, the Table – 1 which deals with age and sex distribution does not give a breakdown of the age groups and refers to a median age alone. 

Response: Thank you for the comment. Our study design was a case control, which was matched by age; therefore, we found most participants were populated in the same age category. However, we have added age breakdown in table1

Comment: The discussion section refers to certain observations which have been compared with other studies and these deal with outcomes in neonates only. 

Response: We appreciate for the comment, we have improved our citation and omitted references which do not match with our study population

Comment: In the section “Materials and Methods “ “study setting and design ‘ the authors have stated that results are obtained in age groups from 1-86 years and of these 50 patients were selected and 150 controls. What was the age distribution taken into consideration when this selection was made. The clinical outcomes may be different in different age groups with polymicrobial bacteremia. 

Response: We appreciate for the good comment, in this study we included all age groups from 0-86. Understanding the age have influence on the clinical outcome, first we adjusted for age in multivariable analysis. Then we did stratification analysis for neonate only and non-neonate. Having this analysis, we are assured we have controlled the age a possible confounder influencing our outcome of interest. 

Comment: The authors have stated a definition for Polymicrobial infections in the section “Definition of terms”. However, the term polymicrobial need not be restricted to 2 organisms as there are instances of blood stream infections caused by more than 2 organisms, including a yeast. How has the selection for polymicrobial infections data been made. Moreover the Results section talks of common organisms as Nos/ 100 which means that the authors have considered only 2 organisms per blood culture in a patient (total no of patients being 50). 

Response: Thank you for this observation. We completely agree polymicrobial infections can be caused by two or more bacteria. Unfortunate, during the review of the laboratory data, we did not come to an instance of having more than two pathogens. Furthermore, we had one case of polymicrobial bloodstream infections involving bacteria and candida species.

Comment: The clinical outcomes for blood stream infections in conditions such as Perforation peritonitis, Carcinoma colon etc may depend on the nos of organisms causing the Polymicrobial Infection. Has this been taken into account. There is no mention anywhere in the manuscript and yeasts do not form part of the study. 

Response: In our analysis we considered co-morbidities as one of the confounders, which could influence our outcome of interest, and in multivariable analysis we controlled for comorbidities, however, we found this did not influence our outcome. Furthermore, we did not document any case with perforation or peritonitis, but there was only one case of polymicrobial infections in patient with rectal carcinoma. In table 2, we have mentioned yeast being part of the study. 

Comment: The authors have stated that Viridans streptococcus and Corynebacteria have been disregarded as contaminants. However, these organisms may assume huge clinical significance in certain clinical situations such as immunocompromise and malignancies in patients with BSI’s. The authors have not provided a break up of clinical conditions in the 50 cases and 150controls which in itself may skew the clinical outcome data of patients. 

Response: Thank you for the comment. Our laboratory SoP for a blood culture states very clearly when to consider CoNS, Viridans Streptococcus, Corynebacterium as true pathogens, and we strictly followed the lab SoP regarding these as pathogenic. Unfortunately, clinical conditions were diverse, and a breakdown presentation in a table would be difficult for a reader.

Comment: The description of Results section under “Antimicrobial Susceptibility Pattern” does not match the data outlined in Table -3. 

Response: Thank you for observation, we have edited on the total number of MDR bacteria

Comment: There is no reference to certain Resistance mechanisms such as Carbapenem resistant Enterobacterales (CRE), Amp C enzymes MLSBi/c detection in S. aureus. All of these do impact treatment outcomes. How have these been factored in the selection of 50 cases and 150 controls in the study as these would most certainly affect the final analyses 

Response: Thank you for the valid comment. In the selection of cases and controls, we did not consider if patients had resistant strains or not because our study hypothesized that polymicrobial bloodstream infections were associated with poor treatment outcomes, and we designed this study as a case control to answer our research question. We had thought of controlling for MDR or resistant strains, but the analysis could have been different and not for a case-control study, maybe cross-sectional.

Comment: The section on Discussion gives vague references to 30-day mortality and extended hospital stay etc without actual figures and how these may compare with similar data brought out in other studies. There are equal references to other studies in neonates without actually discussing the distribution of cases among neonates in the present study. 

Response: Thank you for the comments, we have improved our reference to include reference from similar study group.

Comment: Based on the data presented and discussed the authors may have to modify the conclusion section as no definite conclusions emerge out of the discussion cited. 

Response: Thank you, we have rephrased our conclusion.

---

## [Decision Letter · Decision Letter 2]

27 Mar 2024

Polymicrobial bloodstream infections a risk factor for mortality in neonates at the national hospital, Tanzania: A case-control study

PONE-D-23-28734R2

Dear Dr. Manyahi,

We’re pleased to inform you that your manuscript has been judged scientifically suitable for publication and will be formally accepted for publication once it meets all outstanding technical requirements.

Kind regards,

Iddya Karunasagar

Academic Editor

PLOS ONE

Additional Editor Comments (optional):

All comments have been addressed.

Reviewers' comments:

Reviewer's Responses to Questions

**Comments to the Author**

1. If the authors have adequately addressed your comments raised in a previous round of review and you feel that this manuscript is now acceptable for publication, you may indicate that here to bypass the “Comments to the Author” section, enter your conflict of interest statement in the “Confidential to Editor” section, and submit your "Accept" recommendation.

Reviewer #3: All comments have been addressed

2. Is the manuscript technically sound, and do the data support the conclusions?

Reviewer #3: Yes

3. Has the statistical analysis been performed appropriately and rigorously? 

Reviewer #3: Yes

4. Have the authors made all data underlying the findings in their manuscript fully available?

Reviewer #3: Yes

5. Is the manuscript presented in an intelligible fashion and written in standard English?

Reviewer #3: Yes

6. Review Comments to the Author

Reviewer #3: The authors have addressed the comments and suggestions made by the reviewers adequately and satisfactorily. This manuscript does not appear to be a dual publication, as declared by the authors, and adheres to ethical principles in a laboratory-based study.

7. PLOS authors have the option to publish the peer review history of their article (what does this mean?). If published, this will include your full peer review and any attached files.

Reviewer #3: No

---

## [Editor Report · Acceptance letter]

5 Apr 2024

PONE-D-23-28734R2 

PLOS ONE

Dear Dr. Manyahi, 

I'm pleased to inform you that your manuscript has been deemed suitable for publication in PLOS ONE. Congratulations! Your manuscript is now being handed over to our production team.

Kind regards, 

on behalf of

Dr. Iddya Karunasagar 

Academic Editor

PLOS ONE